# Preventive Effects and Mechanisms of Garlic on Dyslipidemia and Gut Microbiome Dysbiosis

**DOI:** 10.3390/nu11061225

**Published:** 2019-05-29

**Authors:** Keyu Chen, Kun Xie, Zhuying Liu, Yasushi Nakasone, Kozue Sakao, Md. Amzad Hossain, De-Xing Hou

**Affiliations:** 1The United Graduate School of Agricultural Sciences, Kagoshima University, Kagoshima 890-0065, Japan; k4164345@kadai.jp (K.C.); k3591458@kadai.jp (K.X.); liuzhuying@gmail.com (Z.L.); sakaok24@chem.agri.kagoshima-u.ac.jp (K.S.) ; amzad@agr.u-ryukyu.ac.jp (M.A.H.); 2Kenkoukazoku Co., Kagoshima 892-0848, Japan; yasushi.nakasone@kenkoukazoku.co.jp; 3Faculty of Agriculture, University of the Ryukyus, Okinawa 903-0213, Japan; 4Faculty of Agriculture, Kagoshima University, Kagoshima 890-0065, Japan

**Keywords:** garlic, fructan, gut microbiome, 16s RRNA genes, high-fat diet

## Abstract

Garlic (*Allium sativum* L.) contains prebiotic components, fructans, antibacterial compounds, and organosulfur compounds. The complex ingredients of garlic seem to impart a paradoxical result on the gut microbiome. In this study, we used a mouse model to clarify the effects of whole garlic on the gut microbiome. C57BL/6N male mice were fed with or without whole garlic in normal diet (ND) or in high-fat diet (HFD) for 12 weeks. Supplementation with whole garlic attenuated HFD-enhanced ratio of serum GPT/GOT (glutamic-pyruvic transaminase/glutamic-oxaloacetic transaminase), levels of T-Cho (total cholesterol) and LDLs (low-density lipoproteins), and index of homeostatic model assessment for insulin resistance (HOMA-IR), but had no significant effect in the levels of serum HDL-c (high density lipoprotein cholesterol), TG (total triacylglycerol), and glucose. Moreover, garlic supplementation meliorated the HFD-reduced ratio of villus height/crypt depth, cecum weight, and the concentration of cecal organic acids. Finally, gut microbiota characterization by high throughput 16S rRNA gene sequencing revealed that whole garlic supplementation increased the α-diversity of the gut microbiome, especially increasing the relative abundance of f_*Lachnospiraceae* and reducing the relative abundance of g_*Prevotella*. Taken together, our data demonstrated that whole garlic supplementation could meliorate the HFD-induced dyslipidemia and disturbance of gut microbiome.

## 1. Introduction

Garlic (*Allium sativum* L.) has long been used for both culinary and medicinal purposes by many cultures. Based on fresh weight, garlic contains water (62–68%), carbohydrates (26–30%), proteins (1.5–2.1%), amino acids (1.0–0.5%), organosulfur compounds (1.1–3.5%), and fiber (1.5%). Carbohydrates are the most abundant class of compounds present in garlic bulbs and account for about 77% of the dry weight. The majority of the carbohydrate material in garlic consists of water-soluble fructose polymers called fructans [1], accounting for approximately 65% of the dry weight [2]. Garlic fructans are polymerized polysaccharides with high molecular weight ranging from <1000–6800 Da, corresponding to the degree of polymerization [3]. The biological activities of fructans have been intensely investigated as non-digestible polysaccharides or dietary fiber [4,5,6], especially their use as selective substrates to stimulate probiotic bacterial growth and immunomodulation [7,8,9]. Additionally, direct interaction between fructans and intestinal immune cells has been recently suggested [10].

On the other hand, garlic contains 1.1–3.5% organosulfur compounds, which is remarkably higher than that in other plant food. In intact garlic, the primary organosulfur compounds (OSCs) are γ-glutamyl-*S*-allyl-l-cysteines (G-SAC), which are hydrolyzed and oxidized to yield *S*-allyl-l-cysteine sulfoxides (alliin) during storage [11]. Crushing or chopping or chewing garlic releases alliinase, which catalyzes alliin to allicin and other thiosulfates [12]. Allicin is considered to be responsible for most of the pharmacological activity of crushed raw garlic cloves [13]. These OSCs have been thought to be the bioactive principles for numerous health benefits [14], especially for defense components with broad antimicrobial activity.

Intestinal microbes play an important role in maintaining a healthy body [15]. Dietary supplementation with rice bran and navy bean [16], dendrobium polyphenols [17], and propolis [18] has been shown to impact the composition and activities of gut microbiota [19]. High-fat diet (HFD) feeding modulates the gut microbiome composition by decreasing the prevalence of specific gut barrier-protecting bacteria and increasing the prevalence of opportunistic pathogens that can release free antigens such as lipopolysaccharides. This imbalance may be associated with higher gut permeability, leading to higher plasma levels of endotoxin and inflammation factors and eventually the development of metabolic disorders [20,21].

The complex ingredients of garlic seem to have paradoxical results on the gut microbiome. Experiments with separated compounds showed that fructans work as prebiotics for the gut microbiome [22] while garlic OSCs, such as allicin, thiosulfinates, and ajoene, act as antibacterial agents [23,24]. Therefore, it is necessary to clarify the influence of whole garlic intake in daily life on the gut microbiome. In this study, we used a mouse model with normal diet (ND) and HFD to investigate the influence and mechanisms of whole garlic on the gut microbiome. Dextrin was used as positive control because dextrin is a polysaccharide [25], similar to fructan, and can stimulate the growth of probiotic strains such as *Actinobacteria* and *Bacteroidetes* [26], and reduced numbers of pathogenic bacteria [27].

## 2. Materials and Methods

### 2.1. Chemicals and Reagents

Garlic was harvested from a field of soil in Aomori Prefecture, Japan. After hot air drying (moisture content 60%), garlic was stored at −2 °C for 10 months, and then pulverized as garlic crude powder (moisture content 4.8%). The amounts of OSCs and fructan in garlic powder were determined by HPLC or fructan assay kit (Biocon Ltd., Nagoya, Japan), respectively (Appendix A, Table A1).

Indigestible dextrin was obtained from natural corn starch with 95% dextrin and 5% water. Lard oil was obtained from Sigma-Aldrich Japan (Tokyo, Japan). The nutrient composition of the diets is shown in Appendix A, Table A2. ND contained 21% protein, 6% fat, 54% carbohydrate, 4% cellulose, and about 370 kcal/100 g total calories. HFD contained 21% protein, 40% fat, 10% carbohydrate, 4% cellulose, and about 570 kcal/100 g total calories.

### 2.2. Mouse Model

The animal experimental protocol was drafted according to the guidelines of the Animal Care and Use Committee of Kagoshima University (Permission NO. A12005). Male C57BL/6N mice (5 weeks of age) from Japan SLC Inc. (Shizuoka, Japan) were housed separately in cages with wood shavings bedding under controlled light (12-h light/days) and temperature (25 °C), and free access to water and feed. Mice body weight was weighed once a week. After acclimatization for 7 day (6 weeks of age), the mice were randomly divided into six groups (*n* = 5) and fed with ND, NDG (5% garlic in ND), NDD (4% dextrin in ND), HFD, HFDG (5% garlic in HFD), or HFDD (4% dextrin in HFD). After 12 weeks feeding (18 weeks of age), mice were sacrificed after overnight fasting. The fresh feces were collected at the beginning (6 weeks of age) and the end of the experiment (18 weeks of age) for investigating the gut microbiome associated with different ages or diets.

### 2.3. Measurement of Serum Biochemical Indicators

Blood sera were obtained from mice eyeballs and collected in a tube with coagulant (Separable microtubes, FUCHIGAMI, 170720, Kyoto, Japan) for 30 min at room temperature to coagulate properly and were acquired by centrifuging at 3000 rpm for 5 min and stored at −80 °C until use. The serum levels of glutamic-oxaloacetic transaminase (GOT), glutamic-pyruvic transaminase (GPT), gamma-glutamyl transferase (GGT), total cholesterol (T-Cho), total triacylglycerol (TG), high density lipoprotein cholesterol (HDL-c), and glucose were measured with an automated analyzer for clinical chemistry (SPOTCHEM EZ SP-4430, Arkray, Kyoto, Japan). The level of LDLs (low-density lipoproteins) was calculated using the Friedewald equation (LDL = T-Cho − HDL-c − TG/5) [28]. The insulin concentration in serum was measured with an ELISA kit (Thermo Fisher Scientific Inc., Rockford, IL, USA) according to the manufacturer’s instructions. The index of homeostatic model assessment for insulin resistance (HOMA-IR) was calculated with the function of fasting glucose × fasting insulin/405 [29].

### 2.4. Histomorphology

Mice ileum tissue was sliced with a freezing microtome system (Yamato, Saitama, Japan) according to the manufacturer’s instructions. The slice (7 μm) obtained was then stained with hematoxylin-eosin (H&E) staining, and observed under a fluorescence microscope (Keyence, Tokyo, Japan).

### 2.5. Cecal Organic Acid Analyses

Cecum and cecum contents were isolated and weighed. Each 0.3 g sample of cecum content was transferred into 0.6 mL of distilled water, and stood on ice for 10 min after adding 0.09 mL of 12% peroxide acid. The supernatant was filtered after centrifugation with 15,000× *g* at 4 °C for 10 min, and then used for organic acid analysis using ion-exclusion high-performance liquid chromatography with LC-10AD pump (Shimadzu, Kyoto, Japan) and electrical conductivity meter (Waters431, Kyoto, Japan). Component identification was performed by CBM-20A data module (Shimadzu, Kyoto, Japan) [30].

### 2.6. Characterization of the Gut Microbiome by 16S rRNA Gene Sequencing

Mice feces were collected from mice housed in different cages at 6 and 18 weeks age, and stored at −80 °C until use. The fecal genomic DNA was extracted with the Fast DNA spin kit for feces (MP BIOMEDICALS) according to the manufacturer’s instructions, and used for analyzing the composition of gut bacterial communities by sequencing 16S rRNA genes, as described in our previous paper [31].

### 2.7. Statistical Analysis

Results were expressed as mean ± SD or median and range. All of the data were first evaluated with Shapiro-Wilk test to assess the normality of the distribution. The data satisfying the normality were further evaluated using Levene’s test for equal variances to test the equality of variances between populations or factor levels. The data of equal variances were analyzed by one-way analysis of variance (ANOVA) tests, followed by Duncan’s multiple range tests with the SPSS statistical program (version 19.0, IBM Corp., Armonk, NY, USA). A probability of *p* < 0.05 was considered significant.

## 3. Result

### 3.1. Body Weight and Index of Liver Injury

The final body weight of mice fed with HFD at 18 weeks was significantly higher than the mice fed with ND (ND: 36.5 ± 3.34 g, HFD: 47.0 ± 2.19 g, *p* < 0.05 in Figure 1) although there was no difference in initial body weight and in daily food intake throughout the 12-week intervention period. Supplementation with garlic had no significant effect on body weight in both the ND and HFD groups, while supplementation with dextrin increased the body weight in the ND group. Moreover, the serum levels of GOT and GPT were significantly increased (*p* < 0.05 in Figure 2) in the HFD group, and they were significantly reduced (*p* < 0.05) by garlic supplementation. These data indicated the dose of garlic used in this experiment did not result in any damage to the liver and might attenuate the HFD-induced burden of liver, since GPT and the GOT enzymes usually leak out into the general circulation when liver cells are injured. Dextrin, as a polysaccharide control, also showed similar effects on these markers, suggesting that the dose of dextrin used in this study had no any damage to liver.

### 3.2. Effect of Garlic on Metabolism of Lipid and Glucose

To clarify the effect of garlic on metabolism of lipid and glucose, we measured the serum levels of lipid and glucose metabolism markers at the final day of experiment after 12 h fasting. The serum levels of T-Cho, TG, and LDL were significantly increased in HFD group (*p* < 0.05), and they were significantly reduced (*p* < 0.05) by garlic supplementation (*p* < 0.05) (HFDG in Figure 3). Moreover, the serum concentration of insulin was also increased in HFD group, and reduced by garlic supplementation (*p* < 0.05) (HFDG in Figure 4). Although there was no significant difference in serum level of glucose between all groups, the ratio of HOMA-IR, an indicator of insulin resistance, was significantly increased in HFD group, and it was then reduced by garlic supplementation (*p* < 0.05) (HFDG in Figure 4). Dextrin as polysaccharide control showed similar effect on these markers. These data indicated that garlic supplementation attenuated HFD-induced dyslipidemia.

### 3.3. Effect of Garlic on Terminal Ileum Histomorphology and the Concentration of Organic Acids on Cecum

The ratio of villus height/crypt depth is an important indicator for reflecting the digestive and absorptive functions of the small intestine. As shown in Figure 5, HFD decreased the ratio of villus height/crypt depth compared to that in ND (*p* < 0.05). Supplementation with garlic, but not dextrin, significantly recovered the ratio (*p* < 0.05). Moreover, the cecum weight in the HFD group was significantly lower than that in the ND group, and was recovered by garlic or dextrin supplementation (Figure 6A). To further elucidate the effect on the concentration of organic acids in caeca, we measured short-chain fatty acids (SCFAs) and branched-chain fatty acids (BCFAs). The results revealed that the BCFAs, including *iso*-butyric acid, were increased in the HFD group and were reduced by garlic or dextrin supplementation. Acetic acid, propionic acid, n-butyric acid, succinic acid, lactic acid, and formic acid are SCFAs. Of these, the concentrations of butyrate acid and acetate acid were increased in the HFD group and attenuated by garlic or dextrin supplementation. These data indicated that garlic supplementation could attenuate both HFD-induced damage of small intestine morphology and HFD-induced higher concentrations of *iso*-butyric acid, n-butyrate acid, and acetate acid.

### 3.4. Modulation of the Gut Microbiome by Garlic

To further understand the effects of garlic on gut bacteria, the composition and relative abundance of microbiota were determined using high throughput 16 rRNA gene sequencing. The diversity of gut microbiota shown in Figure 7 indicated that supplementation with garlic increased the Chao1 value (A), observed species (B), phylogenetic diversity (PD) whole tree index (C), and Shannon value (D) from 6 to 12 weeks in both the ND and HFD groups. As a control polysaccharide, dextrin decreased all of these four values in the HFD group.

Furthermore, we used principal coordinate analysis (PCoA) plots (β-diversity: between-habitat diversity) based on unweighted UniFrac distance matrices to investigate the similarities in gut microbial community structure among the different groups. The percent of dataset variability explained by each principal coordinate is shown in the axis’s titles (PC1: 15.79%, PC2: 11.60%, PC3: 7.56%). PC1 and PC2 were the two principal coordinate components. PC1 represents the principal coordinate component that can explain the changes in data as much as possible; PC2 represents the principal coordinate component that accounts for the largest proportion of the remaining changes (and so on for PC3). The PCoA plot indicated that the structure of the gut microbiota in the ND group was not obviously changed in aging, but it was altered by HFD. Shifts in the microbial structure were also observed for both garlic and dextrin supplementation, however, there was no significant clustering according to anatomical location. The data suggest that the mechanisms for the regulation of the gut microbiome by garlic and dextrin are different.

Therefore, we further investigated the changes of individual microbial species at the phylum level. The ratio of p_*Firmicutes*/p_*Bacteroidetes* was increased by aging from 6 weeks to 18 weeks in the ND group, and it was attenuated in the NDG and NDD groups (Figure 8). Furthermore, supplementation with garlic increased the relative abundance of f_*Lachnospiraceae,* and decreased the relative abundance of g_*Prevotella* at the genus level of species. In addition, supplementation with dextrin increased the relative abundance of g_*Parabacteroides*, g_*Sutterella*, and f_*Rikenellaceae* (Figure 9).

## 4. Discussion

In this study, garlic supplementation revealed the preventive effects on HFD-induced metabolic disorders and dyslipidemia. These effects included that garlic attenuated HFD-induced increases in LDL serum level, insulin resistance, liver injury, and the concentration of total organic acids in caeca. It has previously been reported that HFD-enhanced the concentration of *n*-butyrate acid, and acetic acid in caecum is associated with the gut microbiome for energy harvest in obese mouse [32]. Butyrate is regarded as the primary energy source for colonic epithelial cells, and propionate and acetate are largely utilized by the liver. Both of them are necessary for lipogenesis and gluconeogenesis [32,33]. Molecular data showed that acetate, propionate, and butyrate can preferentially activate a series of free fatty acid receptors (FFARs) to increase energy harvest and triglyceride storage in adipose tissue [34,35,36]. In addition, we also observed that *iso*-butyrate acid was increased in the HFD group. A raise in *iso*-butyric acid was previously observed in hyperlipidemia, which was positively correlated with an unfavorable lipid metabolism [37]. Therefore, the attenuation of HFD-induced levels of n-butyrate acid, acetic acid, and *iso*-butyrate acid by garlic supplementation might play an important role in the prevention of HFD-induced metabolic disorders and dyslipidemia.

Several lines of studies have stated that moderate consumption of garlic enhanced some gastro-intestinal function, and revealed the protective effect for mucosal defense against *Helicobacter pylori* activity and ulcers development [38,39]. In this study, the jejunal crypt depth was significantly deepened in the HFD group, and garlic supplementation alleviated this situation by promoting epithelial cell renewal and the maturing rate of enterocytes. It is reported that *L*-Glutamate supplementation decreased HFD-deepened crypt depth, and enhanced the cell maturing rate and the secretory function of epithelial cells [40]. On the other hand, excessive consumption of garlic could result in the loss of intestinal epithelial cells [41], leading to the inhibition of intestinal absorption of glutamic acid, sucrose, and glucose [42], which might be the reason why garlic shortens the height of villi in a normal diet.

To understand the effects and mechanisms of whole garlic extract on the gut microbiome, we used a mouse model and compared the data with dextrin, a positive polysaccharide. For this, we further used four different indexes to comprehensively analyze the α-diversity of the gut microbiome. The Chao1 index is a community richness estimator for estimating the number of OTUs (operational taxonomic units) in the sample. The observed species index is a biological species quantitative index, which is calculated according to the number of confirmed OTUs [43,44]. The PD whole tree is a phylogenetic diversity index based on the values of PD. The values of PD are defined as the minimum total length of all the phylogenetic branches on the phylogenetic tree [45]. The Shannon index is a more comprehensive presentation of diversity which is calculated by the scaled OTUs based on community evenness [46]. The results revealed that the β-diversity cluster of microbial communities in the mice supplemented with garlic or dextrin were clustered to different locations, and the supplementation of garlic could enhance the species richness and species evenness of the gut microbiome more than dextrin.

The ratio of p_*Firmicutes* and p_*Bacteroidetes* (F/B) was increased with aging and was reduced in the NDG and NDD groups. Furthermore, the relative abundance of p_*Bacteroidetes;* g_*Prevotella* was increased in the HFD group, but was reduced by garlic supplementation. g_*Prevotella* is reported to induce insulin resistance, aggravate glucose intolerance, and augment the circulating levels of branched-chain amino acids (BCAAs) [47]. The metabolites of BCAAs can further cause insulin resistance by facilitating the transportation of vascular fatty acids [48]. It is noteworthy that the relative abundance of f_*Lachnospiraceae* was upregulated by garlic supplementation in this study. Recent studies demonstrated that f_*Lachnospiraceae* plays an essential role in the maintenance of gut immune homeostasis as the inducer of colonic regulatory T cells [49]. The abundance of f_*Lachnospiraceae* was possibly associated with anti-inflammatory activity [50], and is closely related to host mucosal integrity, bile acid metabolism, polysaccharides decomposition, and protection from colon cancer [51,52]. Moreover, the level of f_*Lachnospiraceae* is correlated negatively with the consumption of energy and positively with the level of leptin [53]. Fructan from garlic has been reported to increase the abundance of f_* Lachnospiraceae* [54,55]. On the other hand, alliin extracted from garlic was found to decrease the abundance of f_*Lachnospiraceae* [56]. These data clearly reveal that fructan and organosulfur derivatives in garlic have opposite effects on the abundance of f_* Lachnospiraceae* when their intakes are isolated from each other. In this study, supplementation with whole garlic, which contained fructan (548 mg/g), alliin (7 mg/g), and other organosulfur derivatives, including allicin (5 mg/g), G-SAC (4 mg/g), and *S*-allylcysteines (SAC, 2 mg/g), could upregulate the abundance of f_*Lachnospiraceae*. Our data revealed that whole garlic could increase the abundance of f_*Lachnospiraceae*, suggesting that the ratio of fructan and organosulfur derivatives is important for gut microbiome. The relative abundance of BCAA-producing bacteria f_* Streptococcaceae* [57], which is significantly increased in patients with cirrhosis [58], was also dramatically increased in the HFD group and was noticeably decreased by garlic and dextrin supplementation. The relative abundance of g_*Akkermansia* was decreased with aging and HFD, and was restored by garlic supplementation. It has been reported that the abundance of g_*Akkermansia* has a negative correlation with the value of body mass index (BMI) [59] and that g_*Akkermansia* has a protective effect for the mucus layer in pro-inflammation [60].

We also observed different effects on some typical bacteria of the gut microbiome from garlic and dextrin supplementation, although both contain polysaccharides. The abundance of g_*Parabacteroides*, g_*Sutterella*, and f_*Rikenellaceae* were increased by dextrin supplementation. The number of g_*Parabacteroides* is enriched by increases in dietary fiber [61], and g_*Parabacteroides* can digest starch that has been chemically modified [62]. g_*Parabacteroides* and f_*Rikenellaceae* are reported to be associated with food allergy mice [63,64]. g_*Sutterella* are widely prevalent commensals with intestinal epithelial cell adhesion and mild pro-inflammatory capacities [65]. These changes in gut microbiome resulting from dextrin supplementation were different than with garlic, and they might be due to the chemical properties of dextrin.

In summary, whole garlic supplementation could attenuate the HFD-induced dyslipidemia and disturbance of the gut microbiome (Figure A1). The data revealed that whole garlic may be a potential prebiotic that is able to prevent against HFD-induced disturbance of the gut microbiome.

## Figures and Tables

**Figure 1 nutrients-11-01225-f001:**
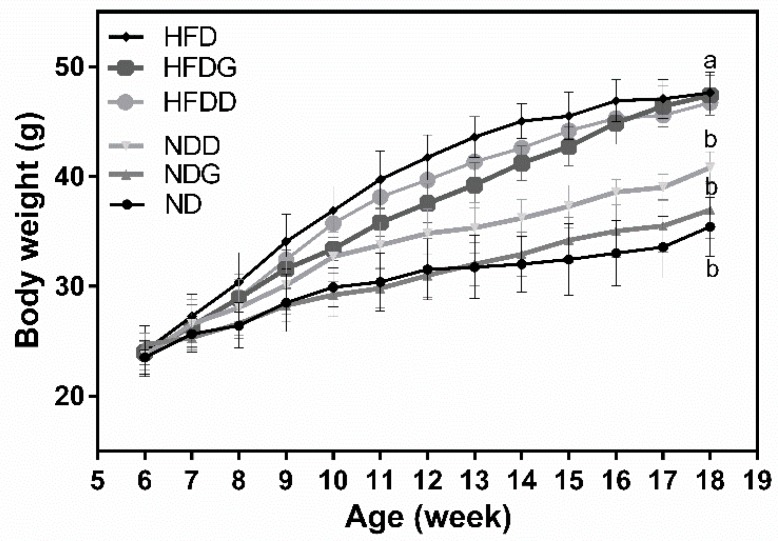
Effects of garlic supplementation on mice body weight. Results are expressed as the mean ± SD for each group of rats (*n* = 5). The body weights at 18 weeks with different letters significantly differ (*p* < 0.05). HFD: high-fat diet; HFDG: 5% garlic in HFD; HFDD: 4% dextrin in HFD; ND: normal diet; NDG: 5% garlic in normal diet; NDD: 4% dextrin in normal diet.

**Figure 2 nutrients-11-01225-f002:**
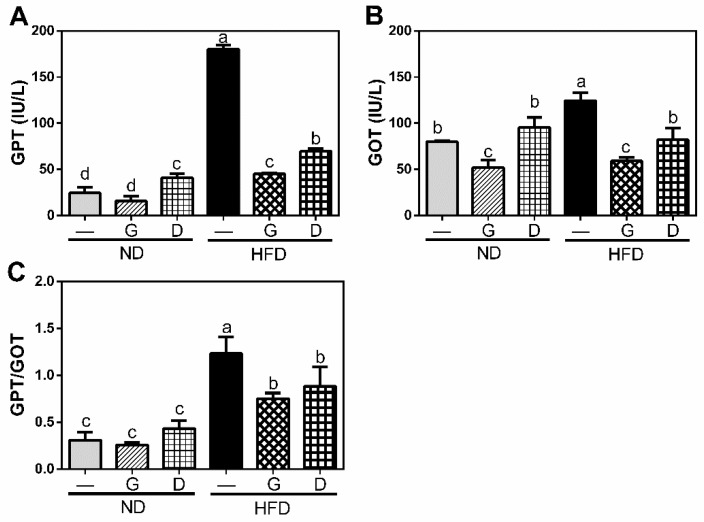
Effects of garlic supplementation on serum level of glutamic-pyruvic transaminase (GPT) (**A**), glutamic-oxaloacetic transaminase (GOT) (**B**), and the ratio of GPT/GOT (**C**). The data represent the mean ± SD of five mice. G: garlic; D: Dextrin. Columns with different letters significantly differ (*p* < 0.05).

**Figure 3 nutrients-11-01225-f003:**
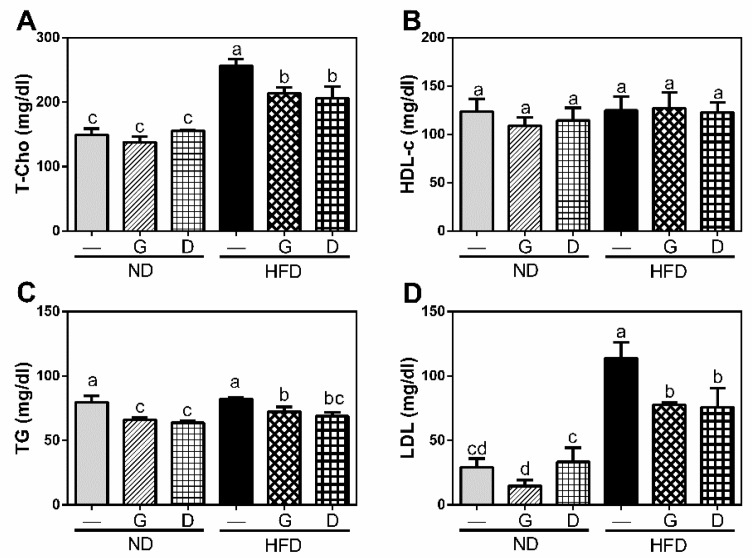
Influence of garlic supplementation on serum level of lipid profiles including total cholesterol (T-Cho) (**A**), high density lipoprotein cholesterol (HDL-c) (**B**), total triacylglycerol (TG) (**C**), and low-density lipoproteins (LDLs) (**D**). The data represent the mean ± SD of five mice for each group. Columns with different letters significantly differ (*p* < 0.05).

**Figure 4 nutrients-11-01225-f004:**
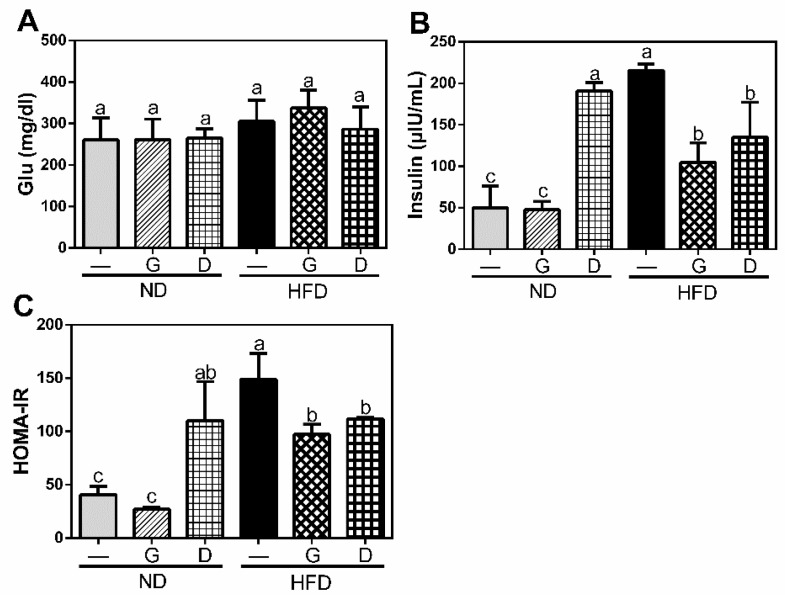
Effects of garlic supplementation on serum level of glucose (**A**) and insulin (**B**) and homeostatic model assessment for insulin resistance (HOMA-IR) (**C**). The data represent the mean ± SD of five mice. Columns with different letters significantly differ (*p* < 0.05).

**Figure 5 nutrients-11-01225-f005:**
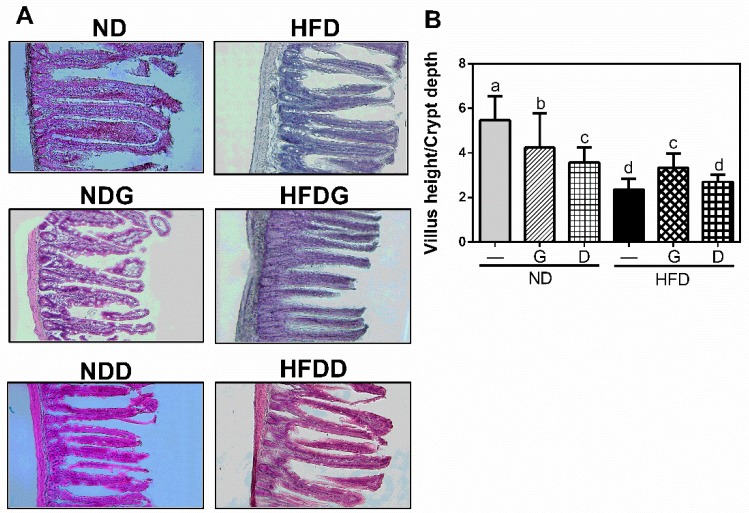
Morphology of terminal ileum (**A**) and the ratio of villus height to crypt depth (**B**). Values are presented as means ± SD of 16 histomorphological points of each group. Columns with different letters significantly differ (*p* < 0.05).

**Figure 6 nutrients-11-01225-f006:**
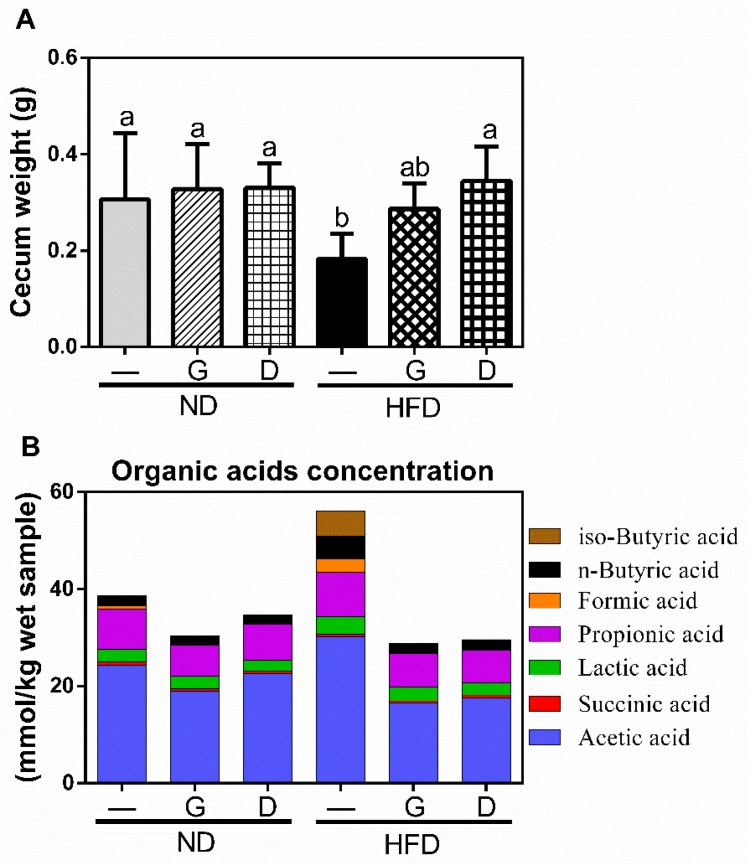
The effects of garlic supplementation on cecum weight (**A**). The cecum weight including cecum and cecum content was measured after the mice were sacrificed. The data represent the mean ± SD of five mice for each group. Columns with different letters significantly differ (*p* < 0.05). The effects of garlic supplementation on organic acid concentration in cecum content (**B**).

**Figure 7 nutrients-11-01225-f007:**
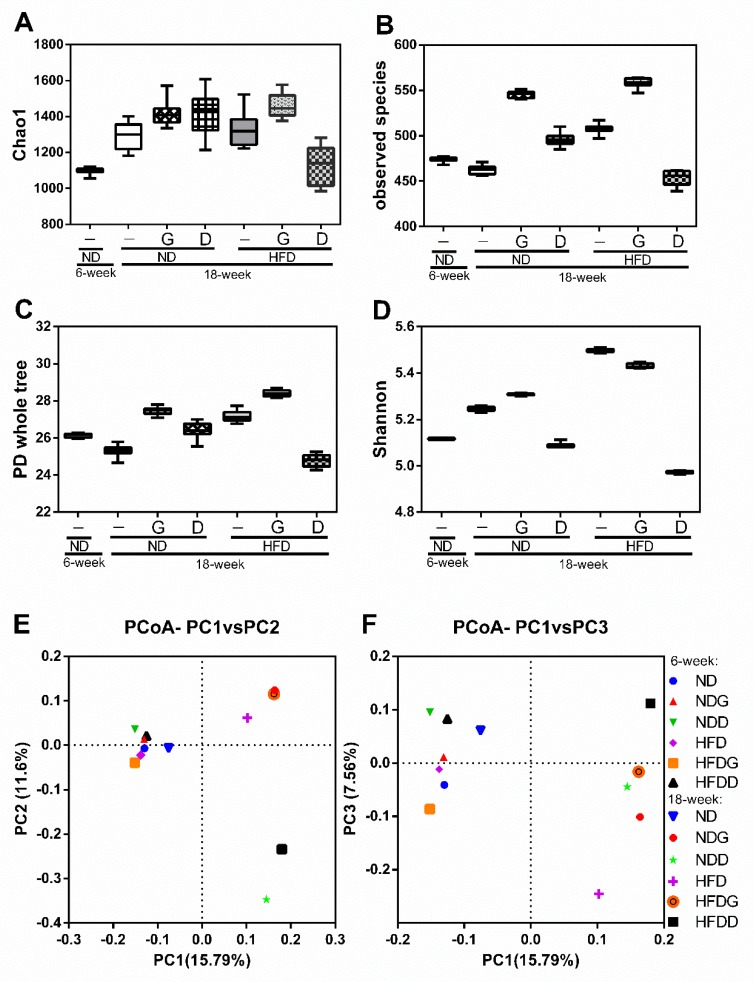
Effects of garlic supplementation on the gut microbiome. The taxa richness of the gut microbiome assessed by α-diversity analyses using Chao1 value (**A**), observed species index (**B**), PD whole tree index (**C**), and Shannon index (**D**). The data represent the median and range of ten alpha rarefaction values. The species compositions of the gut microbiomes were assessed by β-diversity analyses using principal coordinate analysis (PCoA) of the unweighted UniFrac distance matrices, which is showed in PC1 vs. PC2 (**E**) and PC2 vs. PC3 (**F**). Each dot in (E) and (F) represents the beginning (6-week) or ending point (18-week) of the experiment for each group (*n* = 8).

**Figure 8 nutrients-11-01225-f008:**
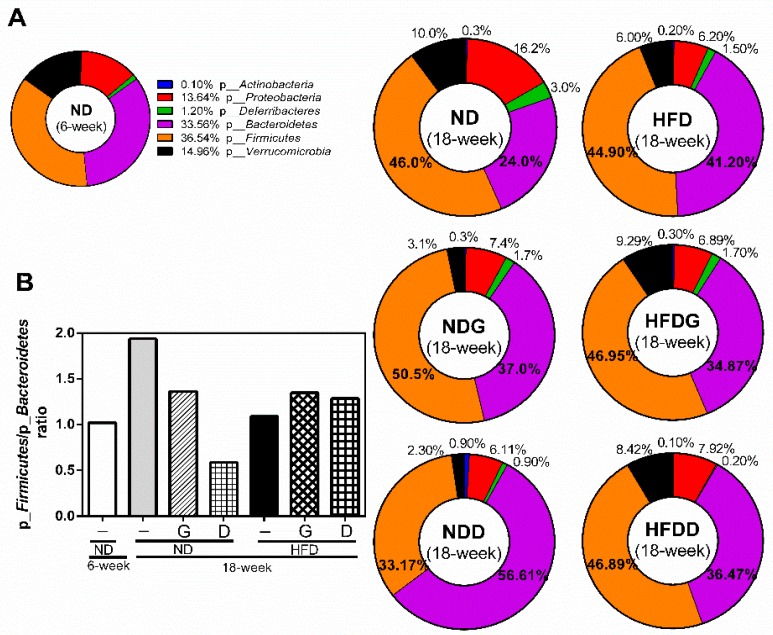
Modulation of the gut microbiome at the phylum level. The gut microbiota was characterized by 16S rRNA gene sequencing. (**A**) The relative abundance of bacteria at the phylum level. (**B**) The ratio of p_*Firmicutes* to p_*Bacteroidetes* based on their relative abundance.

**Figure 9 nutrients-11-01225-f009:**
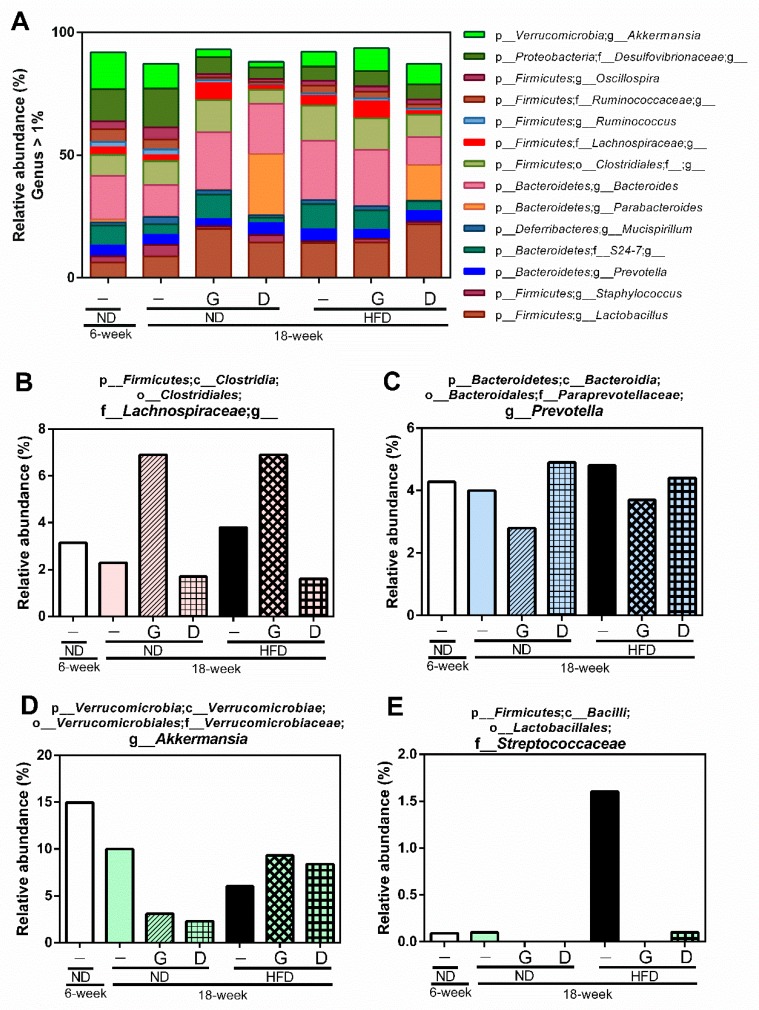
Modulation of the gut microbiome at the genus level. The gut microbiome was characterized by 16S rRNA gene sequencing, and the data represent the relative abundance of each bacterial genus. p_, c_, o_, f_, and g_ represent phylum, class, order, family, and genus, respectively, and a blank after the letter means undefined. (**A**) The relative abundance of more than 1% of bacteria at the genus level. Specifically, four kinds of bacteria (f_*Lachnospiraceae* (**B**), g_*Prevotella* (**C**), g_*Akkermansia* (**D**), f_* Streptococcaceae* (**E**)) were regulated by garlic supplementation.

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
