# Peer review of "Preventive Effects and Mechanisms of Garlic on Dyslipidemia and Gut Microbiome Dysbiosis"

_nutrients, 2019, doi:10.3390/nu11061225_

Round 1

Reviewer 1 Report

Comments to authors

This manuscript described the beneficial effects of garlic on lipids metabolism, intestinal organic acids production which correlated to gut microbiota in a metabolic syndrome mouse model. The study supplies some data on the gut microbiome activities of dietary garlic. Nevertheless, some flaws need to be addressed.

Abstract

Please concise your abstract.

Material and methods

Lines 69 Why the authors select the dextrin as a positive control, explanation on the reasons should be provided clarify.

Lines 77-78 “after hot air drying (moisture content 60%), the samples for experiment were stored at -2℃ for 10 months.” There is a confusion about the moisture content in garlic samples, and in appendix table1, the drying method is described as high pressure drying, do you mean there had another drying procedure after diet preparation? In addition, stored for 10 months is necessary treatment for garlic or not? Moisture content could be a major factor of variation of samples’ nutrition values, please clarify your methods.

Discussion

Lines 245-248 In general, there is a lack of explanation of result α-diversity in the study. an explanation of why the authors did these various statistical analyses should be provided.

Lines 249-250 You mentioned about aging is a factor to increase the ratio of p_Firmicutes and p_Bacteroidetes (F/B ratio), but none of this is described in your methods, If your objective was to compare microbiota composition between ages, then you should explain the age as a factor in your methods.

Lines 284 “The ulcers development of”, of what? Intestine?

Lines 298-299 delete (NDD) and (NDG) or “…reduced in NDG and NDD”.

Lines 325-326 “The trend was similar to the concentration of n-butyrate acid on cecum.” This conclusion is confused, and English is little bit awkward.

Please carefully check your grammar and spellings. There are numerous errors in your text.

Author Response

Response to Reviewer 1 Comments

This manuscript described the beneficial effects of garlic on lipids metabolism, intestinal organic acids production which correlated to gut microbiota in a metabolic syndrome mouse model. The study supplies some data on the gut microbiome activities of dietary garlic. Nevertheless, some flaws need to be addressed.

Abstract

Please concise your abstract.

Response: Yes, we have shortened our abstract and gave more concise abstract in revised manuscript.

Material and methods

Lines 69, why the authors select the dextrin as a positive control, explanation on the reasons should be provided clarify.

Response: Thank you for kind suggestions. Yes, we have added explanations about the prebiotic effect of dextrin to answer why we select the dextrin as a positive control (Lines 68-71).

Lines 77-78 “after hot air drying (moisture content 60%), the samples for experiment were stored at -2 for 10 months.” There is a confusion about the moisture content in garlic samples, and in appendix table1, the drying method is described as high pressure drying, do you mean there had another drying procedure after diet preparation? In addition, stored for 10 months is necessary treatment for garlic or not? Moisture content could be a major factor of variation of samples’ nutrition values, please clarify your methods.

Response: Yes, we had twice drying procedure. The first drying is hot air drying (moisture content 60%) for storage, the second drying is high pressure drying (moisture content 4.8%) for powdering sample. (Lines 75-77). Storage for 10 months is just for applier chain, not necessary treatment for garlic. We have rewritten these in the manuscript.

Discussion

Lines 245-248 In general, there is a lack of explanation of result α-diversity in the study. an explanation of why the authors did these various statistical analyses should be provided.

Response: Yes, we have added explanations about the α-diversity for more comprehensively analysis of species diversity (Lines 301-312).

Lines 249-250 You mentioned about aging is a factor to increase the ratio of p_Firmicutes and p_Bacteroidetes (F/B ratio), but none of this is described in your methods, If your objective was to compare microbiota composition between ages, then you should explain the age as a factor in your methods.

Response: Thank you for your suggestion. We have supplemented the description of the method in 2.2 Mouse model (Line 96-98).

Lines 284 “The ulcers development of”, of what? Intestine?

Response: We have corrected the error. (Line 290).

Lines 298-299 delete (NDD) and (NDG) or “…reduced in NDG and NDD”.

Response: Thank you for the careful reviewing. We have corrected the error. (Line 314).

Lines 325-326 “The trend was similar to the concentration of n-butyrate acid on cecum.” This conclusion is confused, and English is little bit awkward.

Response: We have corrected the error.

Please carefully check your grammar and spellings. There are numerous errors in your text.

Response: Sorry for that, we have checked these throughout the manuscript again.

Finally, we added Dr. Md. Amzad Hossain as a co-author due to his contribution for this work, especial in analysis of garlic components and preparation of the manuscript.

Reviewer 2 Report

This study is very interesting and well-structured. The outcomes are exhaustive and the graphs of results are clear. Nevertheless, one of the limitations of this study could be the limited number of mice for each group (n=5). In this setting, if the normality of data distribution is not verified I suggest to use: 

- median and range (instead of mean and SD) to express results.

- non-parametric statistical test: Kruskal-Wallis (instead of ANOVA) to compare groups medians. 

The graphic abstract is interesting; however, the arrow between alpha-diversity and dyslipidemia is not correctly positioned and should be positioned between HFD diet and dyslipidemia. 

Some English mistakes:

- line 16 replace microbe with microbiome

- line 20-21 [..]but had no significant effect [..]

- line 21- 23: the phrase Moreover [...] cecum is not clear. Please reformulate.

- line 45 has been recently suggested

- line 91 days

- line 143, 154, 164, 165, 178, 183, 190, 205, 211: significantly is not correctly positioned, it should be placed before the verb.

- line 317: whole garlic containing fructans [...] SAC (2mg/g) could up-regulate (invert the periods)

Author Response

Response to Reviewer 2 Comments

This study is very interesting and well-structured. The outcomes are exhaustive and the graphs of results are clear. Nevertheless, one of the limitations of this study could be the limited number of mice for each group (n=5). In this setting, if the normality of data distribution is not verified I suggest to use: 

Response: Thank you for your positive estimation on our paper. We responded your comments one by one as followings.

-median and range (instead of mean and SD) to express results.

Response: Thank you for your careful reviewing. We have corrected the express of results in Fig 7. (Lines 252). By the way, although the number of treated mice was limited to 3-5 mice, we have observed that there was normal distribution after statistical analysis. Thus, we used mean and SD to express the results in other figures to easier understanding. Similar expressions are also used in other articles that their number of treatment mice also had 3-5 mice (Li, J. et al. J. Nutr. Biochem.2017; Wu, S. et al. Mol. Nutr. Food Res. 2017). Thank you for your understanding.

- non-parametric statistical test: Kruskal-Wallis (instead of ANOVA) to compare groups medians. 

Response: Thank you for your suggestion. When the quantitative variables in different groups do not meet the condition of normality, the non-parametric statistical test (Kruskal-Wallis) is often used as an alternative method. We tested the normality by Shapiro-Wilk method. The results show that all of our data satisfied the normality. The added these information and description in 2.7 Statistical Analysis (Line 133-140).

The graphic abstract is interesting; however, the arrow between alpha-diversity and dyslipidemia is not correctly positioned and should be positioned between HFD diet and dyslipidemia. 

Response: Thank you for your comments. We have changed the graphic abstract as you pointed out.

Some English mistakes:

- line 16 replace microbe with microbiome

Response: Yes, we have corrected the error. (Lines 16)

- line 20-21 [..]but had no significant effect [..]

Response: We have corrected the error. (Lines 19)

- line 21- 23: the phrase Moreover [...] cecum is not clear. Please reformulate.

Response: We have corrected the error. (Lines 20-22)

- line 45 has been recently suggested

Response: We have corrected the error. (Lines 44)

- line 91 days

Response: We have corrected the error. (Lines 91)

- line 143, 154, 164, 165, 178, 183, 190, 205, 211: significantly is not correctly positioned, it should be placed before the verb.

Response: Yes, we have moved the position of significantly according to your suggestions in the revised manuscript.

- line 317: whole garlic containing fructans [...] SAC (2mg/g) could up-regulate (invert the periods)

Response: We have corrected the error. (Lines 331-334)

Finally, we added Dr. Md. Amzad Hossain as a co-author due to his contribution for this work, especial in analysis of garlic components and preparation of the manuscript.

Round 2

Reviewer 1 Report

The authors addressed my question adequately. Please make some minor modifications:

1. Line 14- prebiotic components

2. Line 19- abbreviations should be explained

3. Line 123- Serum?

4. Line 343- Helicobacter pylori

Please add following references regarding to foods and microbiota:

https://doi.org/10.1002/mnfr.201500905

https://doi.org/10.1002/mnfr.201800080

https://doi.org/10.3390/molecules23123245

Author Response

Reviewer 1 Review Report (Round 2)

The authors addressed my question adequately. Please make some minor modifications:

1.     Line 14- prebiotic components

Response 1: We have corrected the error (Lines 14), Thank you for your careful reviewing.

2.     Line 19- abbreviations should be explained

Response 2: Yes, we have added full name for the abbreviations. (Lines 19-22)

3.     Line 123- Serum?

Response 3: We could not found “serum” word in line 123. It is supernatant from cecum contents.

4. Line 343- Helicobacter pylori

Response 4: We have corrected the error. (Lines 294).

Please add following references regarding to foods and microbiota:

https://doi.org/10.1002/mnfr.201500905

https://doi.org/10.1002/mnfr.201800080

https://doi.org/10.3390/molecules23123245

Response 4: We have added the references in introduction of dietary supplementation and microbiome. (line 58-60).
